# Relationship between ankle varus moment during gait and radiographic measurements in patients with medial ankle osteoarthritis

**Ji Hye Choi[1], Hee Soo Han[1], Young Jin Park[1], Seungbum Koo[2], Taeyong Lee[3], Kyoung Min Lee[1]** *

1 Department of Orthopaedic Surgery, Seoul National University Bundang Hospital, Seongnam-si, South Korea, 2 Department of Mechanical Engineering, Korea Advanced Institute of Science and Technology, Daejon, South Korea, 3 Division of Mechanical and Biomedical Engineering, College of Engineering, Ewha Womans University, Seoul, South Korea

* oasis100@empal.com

**Data Availability Statement:** All relevant data are within the paper.

**Funding:** This study was funded by Bio & Medical Technology Development Program of the National

## Abstract

### Background

Kinetic data obtained during gait can be used to clarify the biomechanical pathogenesis of osteoarthritis of the lower extremity. This study aimed to investigate the difference in ankle varus moment between the varus angulation and medial translation types of medial ankle osteoarthritis, and to identify the radiographic measurements associated with ankle varus moment.

### Methods

Twenty-four consecutive patients [mean age 65.8 (SD) 8.0 years; 9 men and 15 women] with medial ankle osteoarthritis were included. Fourteen and 10 patients had the varus angulation (tibiotalar tilt angle≥3 degrees) and medial translation (tibiotalar tilt angle<3 degrees) types, respectively. All patients underwent three-dimensional gait analysis, and the maximum varus moment of the ankle was recorded. Radiographic measurement included tibial plafond inclination, tibiotalar tilt angle, talar dome inclination, and lateral talo-first metatarsal angle. Comparison between the two types of medial ankle osteoarthritis and the relationship between the maximum ankle varus moment and radiographic measurements were analyzed.

### Results

The mean tibial plafond inclination, tibiotalar tilt angle, talar dome inclination, lateral talo-first metatarsal angle, and maximum ankle varus moment were 6.4 degrees (SD 3.3 degrees), 5.0 degrees (SD 4.6 degrees), 11.4 degrees (SD 5.2 degrees), -6.5 degrees (SD 11.7 degrees), and 0.185 (SD 0.082) Nm/kg, respectively. The varus angulation type showed a greater maximum ankle varus moment than the medial translation type ($p$ = .005). The lateral talo-first metatarsal angle was significantly associated with the maximum ankle varus moment ($p$ = .041) in the multiple regression analysis.

Research Foundation (NRF) funded by the Ministry of Science & ICT (NRF2017M3A9D8064200).

**Competing interests:** The authors have declared that no competing interests exist.

## Conclusion

The varus angulation type of medial ankle osteoarthritis is considered to be more imbalanced biomechanically than the medial displacement type. The lateral talo-first metatarsal angle, being significantly associated with the ankle varus moment, should be considered for correction during motion-preserving surgeries for medial ankle osteoarthritis to restore the biomechanical balance of the ankle.

## Introduction

Ankle osteoarthritis is a degenerative disease that results in pain and reduced range of motion of the weight-bearing joint, which causes general disability and a reduced quality of life [1]. The prevalence of ankle osteoarthritis has been reported to be 3.4% [2] and is expected to increase with the aging society. Understanding the pathomechanics of ankle arthritis with regard to daily life activities and gait is important because this should provide physicians with more diverse therapeutic approaches to this condition.

Several studies focused on the kinematic and spatiotemporal characteristics of ankle osteoarthritis during gait [3–5]. Decreases in walking speed, stride length, and ankle range of motion were observed in patients with ankle osteoarthritis [5]. Compensatory increases in hip flexion and extension moments were also reported due to decreased plantar flexion moment during the terminal stance in patients with ankle osteoarthritis [6]. However, the results of most of these studies focused on the resultant aspect of the biomechanics, not on the causative biomechanical factors.

The biomechanical pathogenesis of the ankle osteoarthritis is especially important for motion-preserving surgeries such as supramalleolar osteotomy and total ankle replacement arthroplasty because the restoration of biomechanical balance could affect surgical outcomes. A greater preoperative tibiotalar tilt angle (TT) (ankle varus incongruence) was associated with unfavorable outcomes following total ankle replacement arthroplasty or supramalleolar osteotomy [7–11]. However, the biomechanical etiology for these unfavorable outcomes has not been investigated.

Therefore, it has been hypothesized that patients with medial ankle osteoarthritis would have different biomechanics according to TT. Thus, this study aimed to investigate the different ankle varus moments between arthritic ankles with TT $\geq 3$ degrees and those with TT $<3$ degrees, and to analyze the radiographic factors affecting the ankle varus moment.

## Materials and methods

This prospective cohort study was approved by the institutional review board of Seoul National University Bundang Hospital (a tertiary referral medical center), and written informed consent was obtained from each participant.

### Participants

Twenty-four consecutive patients aged $>50$ years, who were diagnosed with medial ankle osteoarthritis on radiographic examination, were enrolled between September 2019 and February 2020. The exclusion criteria were as follows: 1) neuromuscular disease, 2) previous trauma history, 3) infection, 4) tumor, 5) previous foot or ankle surgery, 6) congenital anomaly, 7) inability to walk for any other reason, and 8) any other condition that could change the anatomy of the lower extremity other than ankle osteoarthritis.

## Radiographic examination and measurements

The radiographs were captured using a UT 2000 X-ray machine (Philips Research, Eindhoven, the Netherlands) according to our protocol as follows: the weight-bearing AP view of the ankles was obtained with the horizontal beam centered between the ankle joints at the joint level. The patients were positioned on a 5-cm block, with the film cassette behind the heels. The weight-bearing lateral view of the foot and ankle was captured separately for each foot in the standing position with the beam focusing on the lateral malleolus. The patients were placed in the standing position and the cassette was positioned between both feet. All radiographic images were digitally acquired using a picture archiving and communication system (PACS; Infinitt, Seoul, South Korea), and radiographic measurements were performed using the PACS software.

Four radiographic indices were selected and measured as follows: tibial plafond inclination (TPI) [12], TT [13], talar-dome inclination (TDI) [14], and lateral talo-first metatarsal angle (LTMA) [15]. In the AP view, TPI was measured between the tibial plafond and the horizontal line parallel to the floor [12]. TT was the angle between the tibial plafond and the talar dome [13]. TDI was the angle between the talar dome and the horizontal line (Fig 1A) [14]. In the lateral view, LTMA was measured between the longitudinal axis of the talus and that of the first metatarsal bone (Fig 1B) [15].

To determine interobserver reliability, two orthopedic surgeons with experiences of 5 and 3 years, respectively, performed radiographic measurements for 15 randomly selected patients without the knowledge of the patients' clinical information after consensus was established for measurement. After reliability testing, one surgeon with an experience of 5 years performed radiographic measurements for all the patients.

## Three-dimensional gait analysis

Three-dimensional gait analysis was conducted using a motion analysis system (Motion Analysis Corporation, Santa Rosa, California, USA) equipped with 10 cameras and two force plates.

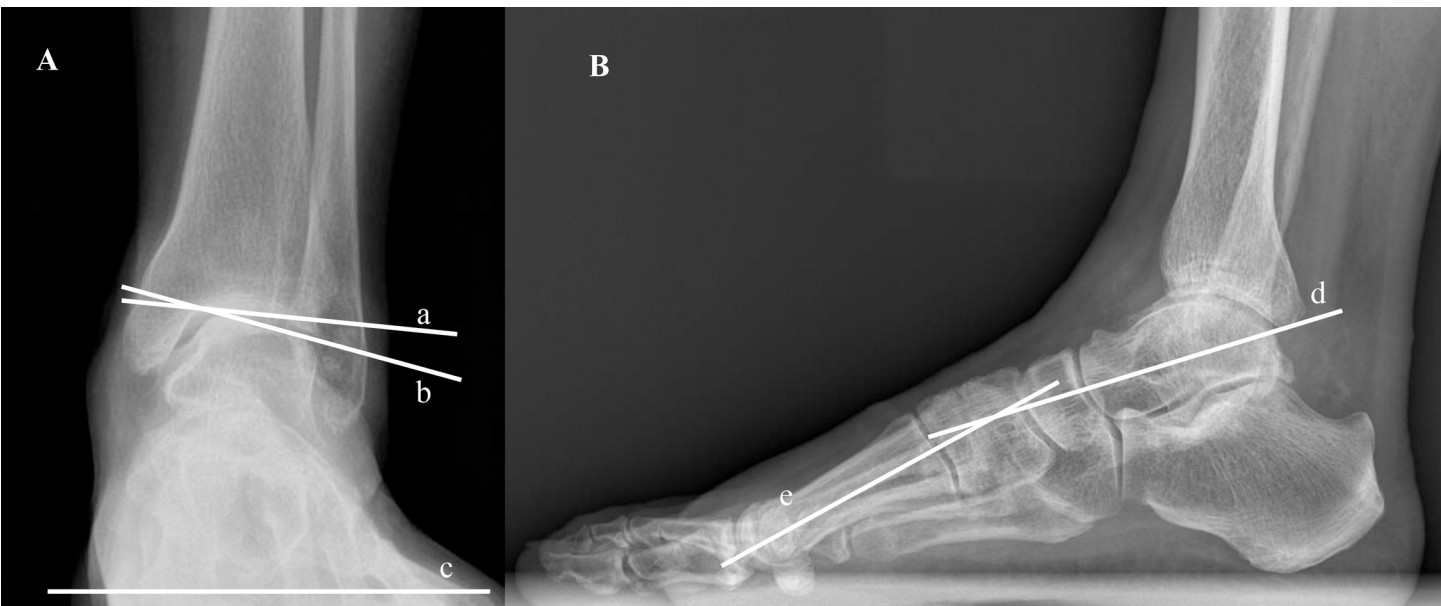

**Fig 1. Radiographic measurements.** A: The ankle in AP view. TPI is the angle between the tibial plafond (a) and the floor (c). TT is measured between the tibial plafond (a) and the talar dome (b). TDI is the angle between the talar dome (b) and the floor (c). B: The foot and ankle in lateral view. LTMA is the angle between the longitudinal axis of the talus (d) and that of the first metatarsal (e).

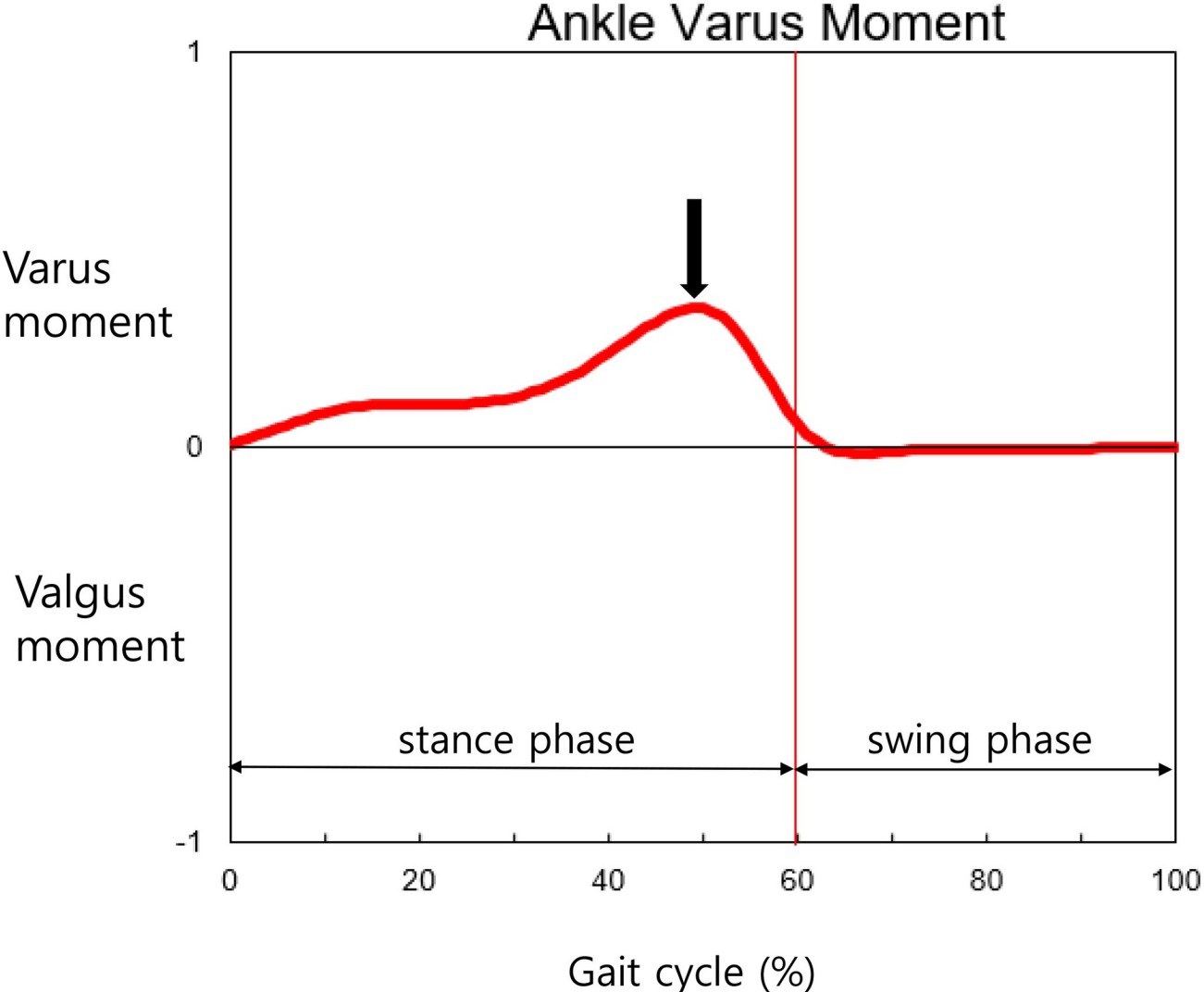

**Fig 2. The maximum angle varus moment (arrow) is the kinetic data in the coronal plane during gait.**

Photo-reflective skin markers were placed according to the Helen Hayes Marker set [16] by a single operator with an experience of 9 years. All participants were instructed to walk bare-footed along a 9-m long track at a self-selected comfortable speed. Three gait trials were selected and averaged to define a gait cycle using the kinematic and kinetic gait variables retrieved for each participant. Of these, the maximum ankle varus moment normalized by body size was retrieved and recorded (Fig 2). Spatiotemporal gait parameters including cadence, step length, and walking velocity were also collected.

### Classification of medial ankle osteoarthritis

Medial ankle osteoarthritis was divided into the varus angulation and medial translation types according to the primary area of joint space narrowing. The tibial plafond and the talar dome tilted with the upper joint space narrowed between the medial talar dome and the medial tibial plafond in the varus angulation type. The medial gutter was primarily narrowed in the medial translation type where the tibial plafond and talar dome were near parallel (Fig 3).

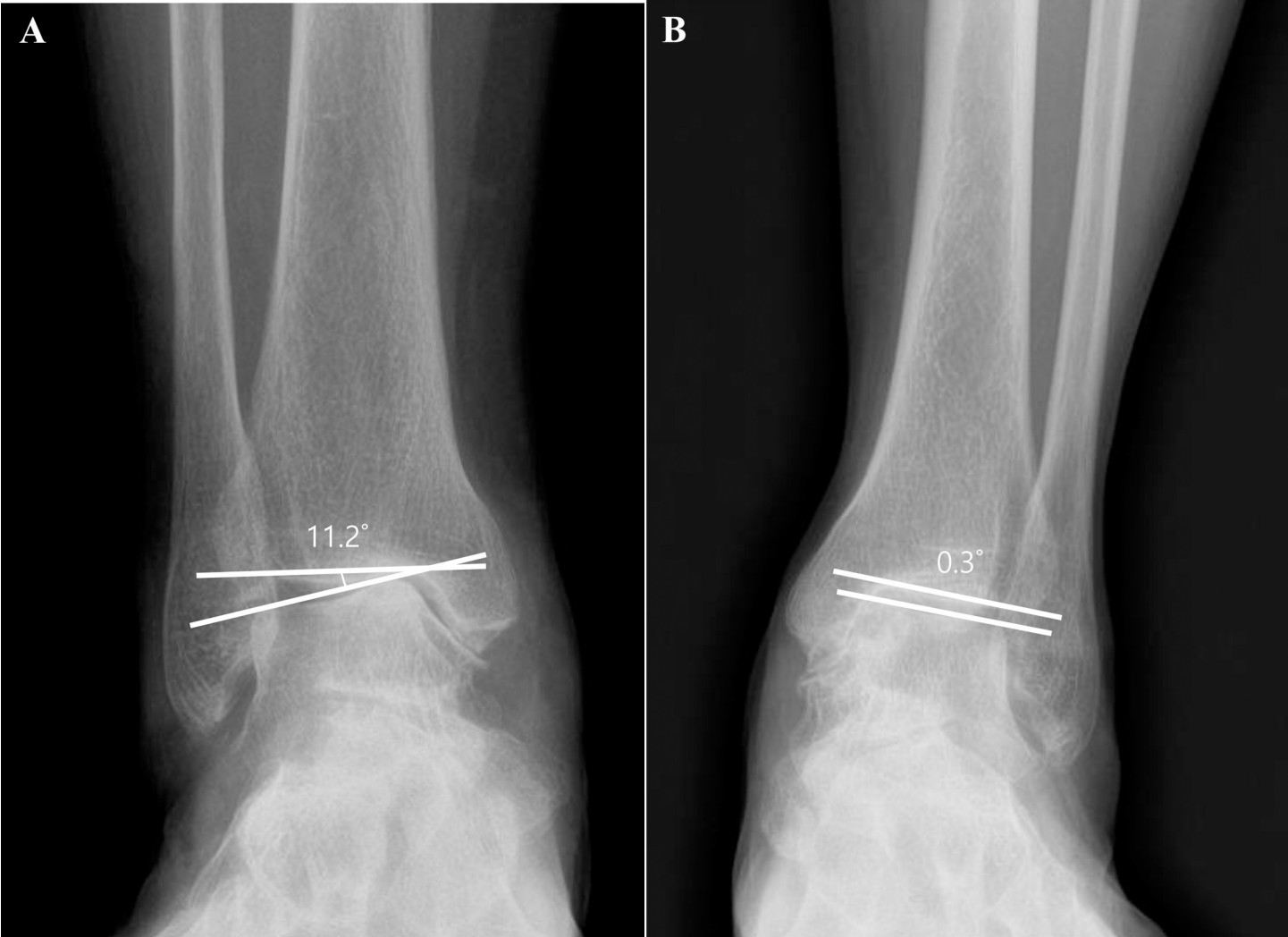

**Fig 3. Two types of medial ankle osteoarthritis.** A: The varus angulation type was defined as TT ≥ 3 degrees. B: The medial translation type was defined as TT < 3 degrees.

Quantitatively, medial ankle osteoarthritis with TT≥3 degrees was classified as the varus angulation type and that with TT<3 degrees as the medial translation type [17].

### Statistical analysis

The results of the descriptive statistical analysis are presented as the average, standard deviation (SD), and proportion. Data normality was confirmed with the Kolmogorov-Smirnov test. Comparison of means between the two types of medial ankle arthritis was performed using the Student t-test or Mann-Whitney U-test as appropriate. Comparison of proportions between the two groups was performed using the chi-square test or Fisher exact test (when the cell data were less than 5). The correlation between the variables was analyzed using Pearson's correlation. Multiple regression analysis was performed to examine the radiographic measurements contributing to the maximum ankle varus moment after univariate analysis; variables with $p \leq$ .1 in the univariate analysis were included in multiple regression analysis. Goodness of fit is presented using adjusted $R^2$ values.

For the interobserver reliability test, the target value of intraclass correlation coefficients (ICCs) for radiographic measurements was 0.9, with a 95% confidence interval (CI) of 0.2. The sample size was calculated, using Bonnett's approximation (ref), as 15 patients for two observers.

All statistical analyses were performed using SPSS version 20.0 (IBM Corp., Armonk, NY, USA), and statistical significance was defined as $p < .05$.

## Results

Twenty-four subjects (9 men, 15 women) with medial ankle osteoarthritis were included in the data analysis. The mean age of the subjects was 65.8 (SD 8.0) years. The mean BMI was 27.3 (SD 4.2) kg/m$^2$. There were 17 right and 7 left arthritic ankles. The mean TT was 5.0 degrees (SD 4.6 degrees) and the mean maximum ankle varus moment was 0.185 (SD 0.082) Nm/kg (Table 1).

All radiographic measurements showed excellent interobserver reliability. LTMA showed the highest interobserver reliability (ICC 0.965; 95% CI, 0.900 to 0.988), followed by TT (ICC 0.958; 95% CI, 0.881 to 0.986), TPI (ICC 0.922; 95% CI, 0.789 to 0.973), and TDI (ICC 0.911; 95% CI, 0.763 to 0.969).

There were no significant differences in age, sex, BMI, and walking velocity between the varus angulation and medial translation types. TT ($p < .001$), LTMA ($p = .001$), and maximum ankle varus moment ($p = .005$) showed significant differences between the two types (Table 2).

TT was significantly correlated with TDI ($r = .636$, $p = .001$) and LTMA ($r = .623$, $p = .001$). TDI showed a significant correlation with LTMA ($r = .481$, $p = .017$). The maximum ankle varus moment was significantly correlated with LTMA ($r = .437$, $p = .033$) (Table 3).

Linear regression analysis showed that LTMA was the only significant radiographic measurement associated with the maximum ankle varus moment ($p = .041$). The explanatory power of the regression model was 25.5% (Table 4).

## Discussion

The results of this study showed that the two types of medial ankle osteoarthritis had different biomechanical characteristics. Patients with a greater TT (ankle varus incongruence) showed a

**Table 1. Data summary.**

| | |
|---|---|
| No. of patients | 24 |
| Age (years) | 65.8 (8.0) |
| Sex (men: women) | 9: 15 |
| Height (cm) | 157.5 (6.7) |
| Weight (kg) | 67.6 (9.1) |
| BMI (kg/m$^2$) | 27.3 (4.2) |
| Side (right: left) | 17: 7 |
| Radiographic measurements | |
| TPI (°) | 6.4 (3.3) |
| TT (°) | 5.0 (4.6) |
| TDI (°) | 11.4 (5.2) |
| LTMA (°) | -6.5 (11.7) |
| Kinetic data | |
| Maximum ankle varus moment (Nm/kg) | 0.185 (0.082) |
| Walking velocity (cm/s) | 92.5 (24.9) |

Values are presented as n or mean (SD).

**Table 2.  Comparison between the varus angulation and medial translation types of medial ankle osteoarthritis.**

|  | Varus angulation type | Medial translation type | p-value |
|---|---|---|---|
| No. of patients | 14 | 10 | - |
| Age (years) | 64.2 (8.5) | 68.0 (6.9) | 0.268 |
| Sex (men: women) | 6: 8 | 3: 7 | 0.418 |
| Height (cm) | 159.6 (6.8) | 154.6 (5.5) | 0.068 |
| Weight (kg) | 68.5 (9.3) | 66.4 (9.2) | 0.592 |
| BMI (kg/m2) | 27.0 (4.7) | 27.8 (3.6) | 0.665 |
| Side (right: left) | 12: 2 | 6: 4 | 0.170 |
| Radiographic measurements |  |  |  |
| TPI (°) | 6.7 (3.1) | 6.0 (3.7) | 0.616 |
| TT (°) | 8.2 (4.0) | 1.5 (1.1) | <0.001 |
| TDI (°) | 14.0 (4.7) | 7.6 (3.2) | 0.001 |
| LTMA (°) | -0.3 (9.3) | -15.2 (9.0) | 0.001 |
| Kinetic data |  |  |  |
| Maximum ankle varus moment (Nm/kg) | 0.22 (0.07) | 0.13 (0.06) | 0.005 |
| Walking velocity (cm/s) | 96.6 (26.4) | 86.7 (22.6) | 0.350 |

Values are presented as n or mean (SD).

greater maximum ankle varus moment during gait, and LTMA (the amount of medial foot arch) was significantly associated with the maximum ankle varus moment in patients with medial ankle osteoarthritis.

The maximum ankle varus moment was significantly different between the two types of medial ankle osteoarthritis. The varus angulation type had greater varus incongruence, as shown by TT, and a greater maximum ankle varus moment during gait than the medial translation type. This implies that the varus angulation type would have greater biomechanical imbalance than the medial translation type. This has been clinically reflected in previous studies that reported unfavorable surgical outcomes following supramalleolar osteotomy or total ankle arthroplasty for ankle osteoarthritis in cases with increased TT [8–11, 18].

Increased TT represents greater varus incongruence at the ankle joint. Increase in the lateral joint space of the ankle could be caused by lateral ligament insufficiency, strong inverter or weak everter muscles, and primary hindfoot varus deformity or secondary hindfoot varus due to an increased medial foot arch (metatarsus primus equinus) [19]. Therefore, patients with medial ankle osteoarthritis with increased TT require thorough examination for these concomitant pathologies, including radiographic and physical examinations such as muscle power, ankle stress tests, and the Coleman block test [18]. The more complex nature of the varus angulation type of medial ankle osteoarthritis is considered to contribute to the greater biomechanical imbalance as shown by the increased ankle varus moment and unfavorable surgical outcome.

**Table 3.  Correlation between radiographic and kinetic variables.**

|  | Maximum angle varus moment | TPI | TT | TDI |
|---|---|---|---|---|
| TPI | 0.368 (p = 0.077) |  |  |  |
| TT | 0.103 (p = 0.632) | -0.172 (p = 421) |  |  |
| TDI | 0.281 (p = 184) | 0.545 (p = 0.006) | 0.636 (p = 0.001) |  |
| LTMA | 0.437 (p = 0.033) | -0.035 (p = 0.872) | 0.623 (p = 0.001) | 0.481 (p = 0.017) |

**Table 4. Radiographic factors significantly associated with the maximum ankle varus moment.**

| | Non-standardized | | Standardized beta | T | p-value |
|---|---|---|---|---|---|
| | B | Standard error | | | |
| TPI | 0.009 | 0.005 | 0.344 | 1.815 | 0.084 |
| LTMA | 0.003 | 0.001 | 0.413 | 2.175 | 0.041 |
| Coefficient | 0.149 | 0.034 | - | 4.402 | <0.001 |

$R^2 = 0.255$.

Our study showed the significant correlation between LTMA and the maximum ankle varus moment. Although TPI did not show a significant correlation with the maximum ankle varus moment, there was a tendency for positive correlation, which could be investigated in future studies with larger sample sizes. This supports the biomechanical basis for supramalleolar osteotomy for medial ankle osteoarthritis because supramalleolar osteotomy directly decreases TPI. Furthermore, for those with a larger TT and greater ankle varus moment, an increased LTMA might need to be corrected with metatarsal osteotomy and/or Dwyer osteotomy [20]. Future clinical studies should investigate biomechanical restoration following those corrective osteotomies.

There are some limitations to be addressed in this study. First, the number of cases was small. Although the important findings were statistically significant, they should be generalized with caution. Second, the compound ankle and hindfoot motion could not discriminate the motions of the ankle and subtalar joints with the Helen-Hayes marker set. Detailed kinematics and kinetics of the ankle and subtalar joints need to be investigated in a future study. Third, the ankle moment obtained from the gait analysis could not discriminate between active and passive moments. The incorporation of EMG data into the gait analysis should help identify pathologic active moment by muscle action and passive moment by abnormal bone or joint alignment.

In conclusion, the varus angulation type of medial ankle osteoarthritis showed a greater maximum ankle varus moment than the medial translation type, suggesting that the former could be associated with more severe biomechanical imbalance than the latter. Our study results showed that LTMA was significantly associated with ankle varus moment, suggesting that LTMA needs to be considered for correction at the time of motion-preserving surgeries for medial ankle osteoarthritis.

## Author Contributions

**Conceptualization:** Kyoung Min Lee.

**Data curation:** Ji Hye Choi, Hee Soo Han.

**Formal analysis:** Ji Hye Choi, Hee Soo Han.

**Investigation:** Seungbum Koo, Taeyong Lee.

**Methodology:** Seungbum Koo, Taeyong Lee.

**Supervision:** Young Jin Park, Seungbum Koo, Kyoung Min Lee.

**Validation:** Hee Soo Han.

**Visualization:** Kyoung Min Lee.

**Writing – original draft:** Ji Hye Choi.

**Writing – review & editing:** Young Jin Park, Kyoung Min Lee.

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
