## [Decision Letter · Decision Letter 0]

9 Jun 2021

Relationship between ankle varus moment during gait and radiographic measurements in patients with medial ankle osteoarthritis

PONE-D-21-13917

Dear Dr. Lee,

We’re pleased to inform you that your manuscript has been judged scientifically suitable for publication and will be formally accepted for publication once it meets all outstanding technical requirements.

Kind regards,

Hans-Peter Simmen, M.D., Professor of Surgery

Academic Editor

PLOS ONE

Additional Editor Comments (optional):

Reviewers' comments:

Reviewer's Responses to Questions

**Comments to the Author**

1. Is the manuscript technically sound, and do the data support the conclusions?

Reviewer #1: Yes

Reviewer #2: Yes

2. Has the statistical analysis been performed appropriately and rigorously? 

Reviewer #1: Yes

Reviewer #2: Yes

3. Have the authors made all data underlying the findings in their manuscript fully available?

Reviewer #1: Yes

Reviewer #2: Yes

4. Is the manuscript presented in an intelligible fashion and written in standard English?

Reviewer #1: Yes

Reviewer #2: Yes

5. Review Comments to the Author

Reviewer #1: It is a well performed study with an interesting, but not completely new information. Still it helps to increase the knowledge about the kinematic action taken into consideration for patient's treatment. The conclusion thus does not give us yet a precise plan in order to decide if the patients with medial ankle OA would benefit of axial alignment surgery or rather be treated by a total ankle replacement. Still the ankle fusion in severe axial deviation might be considered as the safest way to go.

Reviewer #2: This is a well-written and well-conducted study. In addition, the conclusion of it is absolutely correct.

The discussion expands quickly but well over the results and lead to a reasonable interpretation of the results. In addition, the limitations of the study are ok.

6. PLOS authors have the option to publish the peer review history of their article (what does this mean?). If published, this will include your full peer review and any attached files.

Reviewer #1: No

Reviewer #2: **Yes: **Norman Espinosa, MD

---

## [Editor Report · Acceptance letter]

16 Jun 2021

PONE-D-21-13917 

Relationship between ankle varus moment during gait and radiographic measurements in patients with medial ankle osteoarthritis 

Dear Dr. Lee:

I'm pleased to inform you that your manuscript has been deemed suitable for publication in PLOS ONE. Congratulations! Your manuscript is now with our production department. 

Kind regards, 

on behalf of

Dr. Hans-Peter Simmen 

Academic Editor

PLOS ONE